# Utilization of non-pneumatic anti-shock garment and associated factors for postpartum hemorrhage management among obstetric care providers in public health facilities of southern Ethiopia, 2020

**Yordanos Gizachew Yeshitila**[1]*, **Agegnehu Bante**[1], **Zeleke Aschalew**[1], **Bezawit Afework**[2], **Selamawit Gebeyehu**[3]

1 School of Nursing, College of Medicine and Health Science, Arbaminch University, Arba Minch, Ethiopia, 2 Department of Midwifery, College of Medicine and Health Science, Arbaminch University, Arba Minch, Ethiopia, 3 School of Public Health, College of Medicine and Health Science, Arbaminch University, Arba Minch, Ethiopia

* yordanos.gizachew@yahoo.com

## Abstract

### Background

Delays in care have been recognized as a significant contributor to maternal mortality in low-resource settings. The non-pneumatic antishock garment is a low-cost first-aid device that can help women with obstetric haemorrhage survive these delays without long-term adverse effects. Extending professionals skills and the establishment of new technologies in basic healthcare facilities could harvest the enhancements in maternal outcomes necessary to meet the sustainable development goals. Thus, this study aims to assess utilization of non-pneumatic anti-shock garment to control complications of post-partum hemorrhage and associated factors among obstetric care providers in public health institutions of Southern Ethiopia, 2020.

### Methods

A facility-based cross-sectional study was conducted among 412 obstetric health care providers from March 15 –June 30, 2020. A simple random sampling method was used to select the study participants. The data were collected through a pre-tested interviewer-administered questionnaire. A binary logistic regression model was used to identify determinants for the utilization of non-pneumatic antishock garment. STATA version 16 was used for data analysis. A P-value of < 0.05 was used to declare statistical significance.

### Results

Overall, 48.5% (95%CI: 43.73, 53.48%) of the obstetric care providers had utilized Non pneumatic antishock garment for management of complications from postpartum hemorrhage. Training on Non pneumatic antishock garment (AOR = 2.92; 95% CI: 1.74, 4.92),

**Data Availability Statement:** All relevant data are within the paper and its Supporting information files.

**Funding:** This study was funded by Arba Minch University as a part of a project with a grant code of GOV/AMU/TH12/CMHS/NUR/02/11. The website of the university is www.amu.edu.et. The funders had no role in study design, data collection, and analysis, decision to publish, or preparation of the manuscript.

**Competing interests:** The authors have declared that no competing interests exist.

working at hospital (AOR = 1.81; 95% CI: 1.04, 3.16), good knowledge about NASG (AOR = 1.997; 95%CI: 1.16, 3.42) and disagreed and neutral attitude on Non pneumatic antishock garment (AOR = 0.41; 95%CI: 0.24, 0.68), and (AOR = 0.39; 95% CI: 0.21, 0.73), respectively were significantly associated with obstetric care provider's utilization of Non-pneumatic antishock garment.

## Conclusions

In the current study, roughly half of the providers are using Non-pneumatic antishock garment for preventing complications from postpartum hemorrhage. Strategies and program initiatives should focus on strengthening in-service and continuous professional development training, thereby filling the knowledge and attitude gap among obstetric care providers. Health centers should be targeted in future programs for accessibility and utilization of non-pneumatic antishock garment.

## Introduction

Postpartum hemorrhage (PPH) is a substantial contributor to severe maternal morbidity, long-term complications, and disabilities as well as to several other severe maternal conditions commonly associated with more considerable blood loss, including shock and organ dysfunction. A woman suffering from PPH may die within two hours unless she receives immediate and prompt medical care [1, 2]. Globally, postpartum hemorrhage accounts for one-quarter of all maternal death and it is the leading cause of maternal mortality in low-income countries [1]. In Ethiopia, about 25% of maternal death are attributed to hemorrhage [3, 4].

The survival and health of both mother and fetus depend on the ability of families and communities to recognize and access care quickly in case of an emergency. For obstetric complications like hemorrhage, the window of opportunity to respond and save the life of the mother may be measured in hours [3]. Delays in receiving care have been recognized as major contributors to maternal mortality in low-resource settings. The non-pneumatic anti-shock garment (NASG) is a low-cost first-aid device that may help women with obstetrical hemorrhage survive these delays without long-lasting adverse effects [5–7].

The NASG is a simple neoprene and velcro device that looks like the bottom half of a wetsuit cut into segments. It can be used to treat resuscitate, stabilize and prevent further bleeding in women with obstetric hemorrhage. It comprises nine articulated segments that are wrapped sequentially around the legs, pelvis, and abdomen and closed with velcro. A foam ball over the abdomen provides additional compression. The NASG works by applying circumferential counter pressure, decreasing blood flow to the compressed area (abdomen, pelvis, and lower extremities), and enhancing blood flow to the heart, lungs, and brain. As blood flow increases to these core organs, the symptoms of shock are reversed [8].

The NASG can be applied by any healthcare professional after brief training, and it results in the reversal of hypovolemic shock stabilization of the patient for many hours, during transport, examinations, and delays in receiving definitive treatments such as blood, procedures, and surgeries. The NASG is the only device available to stabilize women with shock until definitive treatments can be given. If Intravenous (IV) fluids no longer flow, the veins are easier to find after placing the garment. Because hypovolemic patients may not have access to surgery and/or blood, the NASG can help maintain patients while awaiting definitive care [9].

Even with major advances in the prevention of PPH women are still dying [10]. The use of non-pneumatic anti-shock garments is recommended as a temporizing measure until appropriate care is available. The World Health Organization (WHO) recommends health facilities delivering maternity services should adopt formal protocols for the prevention and treatment of PPH and patient referral [1]. Clinton Health Access Initiative (CHAI) introduced the non-pneumatic anti-shock garment (NASG) in selected health facilities of Ethiopia in June 2011 to reduce complications due to pregnancy-related excessive bleeding. With the introduction of the NASG, there was a 79% reduction in maternal deaths due to postpartum hemorrhage [11].

Before and after studies conducted in different parts of the world revealed a 55% reduction in maternal mortality, 80% reduction in blood loss, emergency hysterectomy decreased from 8.9% to 4.0%, and severe adverse outcomes lessened from 12.8% to 4.1% due to the application of NASG [12–15]. Evidence-based information from randomized control trials also suggested faster recovery from obstetric shock, decreased maternal mortality, severe end-organ failure, and morbidity after the utilization of NASG [16–18]. The NASG is also suggested to have major implications for nurses and midwives attending or assisting in childbirth in low resource settings with delays in obtaining definitive therapy [9].

Postpartum hemorrhage is one of the most alarming and serious emergencies which health professionals may face at health centers and their prompt and competent action will be crucial in controlling blood loss and reducing the risk of maternal morbidity or even death by using low cost, effective, and lifesaving instrument (NASG). Most of the studies conducted before used a small sample size (60–112) [19–22]. Non-availability of NASG, working experience, age of participants, qualification, lack of skilled personnel, and not being aware of the existence of NASG were factors identified in the previous studies about knowledge and utilization of NASG [20, 22–25].

Notwithstanding the benefit of NASG for preventing complications of obstetric hemorrhage in low-income countries like Ethiopia, where almost all maternal mortality occurs and for which 25–28% of maternal mortalities are attributable to postpartum hemorrhage few studies have been conducted to assess the knowledge and practice of health professionals about the use of NASG. Thus this study aimed to assess obstetric care providers utilization of NASG for the prevention of complications from postpartum hemorrhage and associated factors in public health institutions of southwest Ethiopia.

## Materials and methods

### Setting and duration of the study

This study was conducted in public health facilities of Gamo, Gofa, Segen Areas People, and South Omo Zone, Ethiopia from March 15 –June 30, 2020. Gamo, Gofa, Segen Areas People, and South Omo area are administrative Zones in the Southern part of Ethiopia. Those Zones hosted different general and district hospitals that serve the community by providing preventive and curative services. There are five functional hospitals in Gamo Zone (Arba Minch General Hospital, Chencha District Hospital, Kamba District Hospital, Gerese District Hospital, and Selamber District Hospital), one hospital in Gofa Zone (Sawla District Hospital), two hospitals in Segen Areas People Zone (Karat District Hospital and Gidole District Hospital) and three hospitals in South Omo Zone (Jinka General Hospital and Gazer District Hospital). Regarding the health centers distribution, there are a total of 76 in Gamo zone, 56 in Gofa zone, 16 in Segen area, 12 in Konos zone, and 41 in south Omo.

### Study design and population

Institution-based cross-sectional study design was employed among obstetric care providers. All health care professionals who were working in public health facilities of southwest Ethiopia

were the source population. And all obstetric care providers who were working in public health facilities of southwest Ethiopia, where NASG is available for PPH management and fulfilled the selection criteria were the study population.

### Inclusion and exclusion criteria

All obstetric care providers who were staff in the respective wards in each public health facility were included in this study, whereas those obstetric care providers on annual leave at the time of data collection were excluded from this study.

### Sample size and sampling procedure

A single population proportion formula was used to estimate the sample size required for the study. The sample size calculation assumed the proportion (p), the estimated level of utilization of non-pneumatic antishock garment 52.2% [23], 95% confidence level, and margin of error of 5% which gave a sample size of 384. In consideration of a 10% non-response rate, the final sample size was 422 obstetric care providers. There are eleven fully functional hospitals in five Zones of Southern Ethiopia. All the hospitals were included in the study. Regarding the health centers, among the 113 health centers in the five zones, 25% (34) of the health centers were selected by a simple random sampling method. At first, the calculated sample size was proportionally allocated to each public health facility based on the number of obstetric care providers who were working in the respective facilities. Furthermore, a table of the random number was used to select each obstetric care provider based on proportions to get the desired sample size.

### Data collection instruments and procedures

A self-administered semi-structured questionnaire was used to collect data from study participants. The data collection tool is developed by reviewing different kinds of literature and guidelines [8, 19–24, 26–32] and it consists of eight parts which include: socio-demographic characteristics, professional related characteristics, facility-related characteristics, contextual related characteristics, and health professional's knowledge about NASG, health professional's utilization of NASG, the attitude of health care professional NASG which had eight questions and the responses consist of five Likert scales which were strongly agree, agree, neutral, disagree, and strongly disagree.

### Definitions and measurements

**Utilization of non-pneumatic anti-shock garment.** Measured based on the response to the question of whether the obstetric care providers used NASG for the management of postpartum hemorrhage at least one time [23].

**Knowledge scale.** The respondents' score of total knowledge questions those who below 50% were graded as having poor knowledge while those who score greater than 50% from the provided knowledge questions were graded as having good knowledge [33].

**Attitude scale.** A Likert scale was created by presenting respondents with a series of negative and positive attitude statements with five possible responses. For negative statements, responses including agree, and, strongly agree, were labeled as "disagree", disagree and strongly disagree were labeled as "agree", and undecided responses were labeled as "neutral responses".

## Data processing and analysis

Data were analyzed using STATA version 16. Data were cleaned by running frequencies to each variable to check outliers, inconsistencies, and missed values. The outcome variable was utilization; those who utilized NASG were coded as "1" and those who did not utilize were coded as "0". The assumptions for binary logistic regression were checked. Hosmer-Lemeshow statistic and Omnibus tests were done for model fitness. Variables with P < 0.25 in the bivariate analysis, and variables that were significant in previous studies were considered to select the candidate variables for the final model. Collinearity statistics (Variance inflation factor (VIF) > 10 and tolerance (T) < 0.1 were considered as suggestive of the existence of multi co-linearity. Adjusted Odds Ratio along with 95% CI was estimated to identify factors affecting obstetric care provider's utilization of NASG. The P-value < 0.05 was considered to declare a result as statistically significant. Then simple frequencies, summary measures, tables, and figures were used to present the information.

## Data quality management

Data collectors and supervisors were provided with a daylong intensive training on the techniques of data collection and components of the data collection tool. Before the actual data collection, the questionnaire was pre-tested on 10% of obstetric care providers in Wolayita Sodo Hospitals. Based on the findings from the pretest, ambiguous questions were amended. An ongoing formative checkup for completeness and consistency of responses was made by the supervisors daily.

## Ethics consideration and consent to participate

Ethical clearance was issued from the Institutional Review Board of Arba Minch University. Permission was secured from the respective hospital and health center administrators. Moreover, written consent was obtained from each study participant before the commencement of data collection. Before obtaining the consent of each participant, a letter of support and approval to undertake the research in health facilities was obtained from managers in each public health facility. Privacy, as well as the confidentiality of participants, was asserted. In any case, their right to withdraw from the study at any time was assured.

# Results

## Socio-demographic characteristics of the study participants

A total of 412 obstetric care providers participated, yielding a response rate of 97.6%. The mean age and standard deviation of study participants were 27.77 ± SD 3.98 years old. Two hundred ninety-four (71.36%) of the obstetric care providers reside in urban areas (Table 1).

Out of the total respondents, 298 (71.12%) of the obstetric care providers were working in health centers. Of the total obstetric care providers, 150 (36.4%) had training about NASG. Two hundred fifty-four, (61.6%) of the obstetric care providers had protocols about the NASG in their respective facilities (Table 2).

## Participants' knowledge of non-pneumatic anti-shock garment

This study indicated that the majority (82.04%) of respondents had heard about NASG while 74 (17.96%) never heard it before. One hundred sixty-one (47.6%) of the respondents correctly mentioned that NASG has six parts. Regarding the type of pressure, NASG exerts when applied on a patient, half of the respondents correctly mentioned that circumferential pressure is applied. Regarding the contra-indication to the use of NASG, the majority (83.73%) of the

**Table 1. Socio-demographic characteristics of obstetric care providers who participated in the study and were working in public health facilities of southern Ethiopia, 2020 (N = 412).**

| Variable | Frequency | Percent |
|---|---|---|
| **Age in years** | | |
| 20–24 | 81 | 19.7 |
| 25–29 | 206 | 50 |
| 30–34 | 92 | 22.3 |
| 35–40 | 33 | 8 |
| **Sex** | | |
| Female | 268 | 65 |
| Male | 144 | 35 |
| **Marital status** | | |
| Married | 255 | 61.9 |
| Single | 148 | 35.9 |
| Others * | 9 | 2.2 |
| **Religious status** | | |
| Protestant | 195 | 47.3 |
| Orthodox | 195 | 47.3 |
| Muslim | 16 | 3.9 |
| Catholic | 6 | 1.46 |
| **Residency** | | |
| Urban | 294 | 71.4 |
| Rural | 118 | 28.6 |

Others * = divorced, married but living apart.

respondents responded it shouldn't be applied to the viable fetus (Table 3). Overall, about three-fourth (74.26%) of the obstetric care providers had good knowledge about NASG.

## Attitude about non-pneumatic anti-shock garment

Regarding the attitude of the respondents towards NASG, one hundred ninety-eight (48%) of the respondents disagreed that the use of anti-shock garment (NASG) is unnecessary especially in the center where there is a facility for blood transfusion. Around two-fifth, (39%) of the respondents agreed that the garment (NASG) is expensive, therefore not affordable (Table 4).

## Utilization of non-pneumatic anti-shock garment

Regarding the utilization of NASG by obstetric care providers, two hundred (48.4.2%) used NASG for postpartum hemorrhage management while 51.45% never applied NASG while managing a patient with postpartum hemorrhage. More than half (54%) of the respondents mentioned "Don't know how to use it" as the reason for non-utilization (Table 5).

Overall, 48.4% (95% CI: 43.7, 53.4%) of the study participants had NASG for management of postpartum hemorrhage.

## Factors associated with utilization of non-pneumatic antishock garment

Training on NASG, type of facility, attitude towards NASG, and good knowledge on NASG was significantly associated with obstetric care provider's utilization of NASG after controlling for confounders in the multivariable model.

**Table 2. Provider and facility-related factors of the obstetric care providers in public health facilities of Southern Ethiopia, 2020.**

| Variable | Frequency | Percent |
|---|---|---|
| **Profession** | | |
| Midwifes | 304 | 73.8 |
| Nurses | 51 | 12.4 |
| Medical doctor | 9 | 2.2 |
| Integrated Emergency Obstetric Surgery | 48 | 11.6 |
| **Educational level** | | |
| Diploma | 235 | 57 |
| Bachelor degree | 120 | 29.1 |
| Masters | 57 | 13.8 |
| **Years of experience** | | |
| 1–5 years | 241 | 58.5 |
| 6–10 years | 145 | 35.2 |
| 11 years and above | 26 | 6.3 |
| **Unit of service** | | |
| ANC | 79 | 19.2 |
| Family planing | 22 | 5.3 |
| Labour and delivery | 210 | 60 |
| GYN Opd | 63 | 15.3 |
| Postnatal | 38 | 9.2 |
| **Facility type** | | |
| Hospital | 119 | 28.9 |
| Health center | 293 | 71.1 |
| **Protocol about NASG in the facility** | | |
| Yes | 254 | 61.6 |
| No | 158 | 38.3 |
| **Traning about NASG** | | |
| Yes | 150 | 36.4 |
| No | 262 | 63.6 |

Obstetric care providers who received training on NASG were 3 times more likely to use NASG as compared to those who had no training (AOR = 2.92,95%CI: 1.74, 4.92).

Obstetric care providers who disagreed, and those who had a neutral attitude about NASG were 59% and 61% less likely to use NASG as compared to those who had a positive attitude, (AOR = 0.41, 95% CI: 0.24, 0.68) and (AOR = 0.39, 95CI: 0.21,0.73) respectively. The odds of the utilization of NASG among obstetric care providers who are working at hospitals were 1.81 times more likely to use NASG as compared to those who work at health centers (AOR = 1.81, 95%CI: 1.04, 3.16). Those who had good knowledge about NASG were 1.99 times more likely to use NASG (AOR = 1.99, 95% CI; 1.16, 3.42) (Table 6).

## Discussion

In this study, 48.5% (95%CI: 43.73, 53.48%) of the obstetric care providers had utilized NASG for the management of complications from postpartum hemorrhage. Training on NASG, type of facility, knowledge, and attitude on NASG were significantly associated with obstetric care provider's utilization of NASG.

**Table 3. Knowledge of NASG among obstetric care providers in public health facilities of southern Ethiopia, 2020.**

| Variables | Frequency | Percentage |
|---|---|---|
| **Heard of NASG (N = 412)** | | |
| Yes | 338 | 82 |
| No | 74 | 18 |
| **Knew NASG as it is used for preventing complications from PPH (N = 338)** | | |
| Yes | 254 | 75.1 |
| No | 84 | 24.9 |
| **Source of information (N = 338)** | | |
| Health institution | 265 | 78.4 |
| School | 76 | 22.5 |
| Seminars | 30 | 8.9 |
| Internet | 17 | 5 |
| Printed material | 5 | 1.5 |
| Friends/colleagues | 15 | 4 |
| **NASG looks like (N = 338)** | | |
| *Bottom half of a suit | 150 | 44.4 |
| A gown | 30 | 8.9 |
| A trouser | 158 | 46.8 |
| **NASG is made of (N = 338)** | | |
| *Velcro | 66 | 19.5 |
| *Neoprene | 78 | 23.1 |
| I don't know | 194 | 57.4 |
| **Number of segments of NASG (N = 338)** | | |
| Four | 130 | 38.5 |
| *Six | 161 | 47.6 |
| Nine | 17 | 5 |
| Five | 30 | 8.9 |
| **Type of pressure NASG exerts when applied on a patient (N = 338)** | | |
| *Circumferential | 169 | 50.1 |
| Direct | 132 | 39.2 |
| Counter | 36 | 10.7 |
| ***Types of activities performed on a woman on Non-Pneumatic Antishock Garment© (N = 338)** | | |
| Intravenous line | 268 | 79.3 |
| Vaginal surgery | 173 | 51.2 |
| Abdominal surgery | 147 | 43.5 |
| Transport to other facilities | 235 | 69.5 |
| ***NASG removed when© (N = 338)** | | |
| After stabilizing for 2hrs | 212 | 62.7 |
| When the hg is 7g/ dl or more and hematocrit of about 20% | 179 | 53 |
| Pulse rate less than 100 bpm | 167 | 49.4 |
| Diastolic bp 90mmhg or more | 156 | 46.2 |
| When the women is awake /stable | 133 | 39.4 |
| Bleeding <50ml/hr | 111 | 32.8 |
| **Starting segment while applying NASG (N = 338)** | | |
| Abdominal segment | 60 | 17.8 |
| * Lower or ankle segment | 254 | 75.2 |
| I don't know | 24 | 7.1 |

*(Continued)*

**Table 3.** (Continued)

| Variables | Frequency | Percentage |
|---|---|---|
| **Starting segment while removing NASG (N = 338)** | | |
| Abdominal segment | 82 | 24.3 |
| *Lower or ankle segment | 219 | 64.8 |
| I don't know | 37 | 10.9 |
| **Time interval to remove each segment (N = 338)** | | |
| Each successively | 93 | 27.5 |
| * 15 minutes apart | 177 | 52.4 |
| 1hr apart | 44 | 13 |
| I don't know | 24 | 7.1 |
| **Segment adjusted when a woman experiences difficulty breathing with the NASG (N = 338)** | | |
| *Abdominal | 229 | 67.8 |
| Thigh | 44 | 13 |
| leg | 39 | 11.5 |
| I don't know | 26 | 7.7 |
| **Contra-indication to the use of NASG© (N = 338) | | |
| Viable fetus | 283 | 83.7 |
| Dyspnea | 250 | 74 |
| Mitral stenosis | 223 | 66 |
| CHF | 237 | 70.1 |
| Pulmonary hypertension | 227 | 67.2 |
| Bleeding above diaphragm | 230 | 68 |
| **How long the NASG can/should be used on a given patient (N = 338)** | | |
| For two hours | 120 | 35.5 |
| For 48 hours | 65 | 19.2 |
| * Applied until the bleeding arrested | 135 | 39.9 |
| I don't know | 18 | 5.3 |
| **Knowledge** | | |
| Good knowledge | 303 | 74.3 |
| Poor knowledge | 105 | 25.7 |

©Multiple responses are possible.

*Correct response,

** all the options are the responses, hg: hemoglobin, bp: blood pressure, bbp: beat per minute, mmhg: millimeter of mercury.

The magnitude of the utilization of NASG among obstetric care in the current study is in line with a study done in western Nigeria (52.2%) and higher than studies conducted in Jimma, Ethiopia (36.2%), Ibadan Nigeria (35%), Ondo-State, Nigeria (14.1%) [21, 23, 33, 34]. The possible explanation for this discrepancy could be due to a large number of public health facilities (11 hospitals and 34 health centers) included in the current study. Whereas the magnitude of the current study is lower than the study conducted in the northern part of Ethiopia [35]. This could be because the introduction of the instrument NASG in Ethiopia started in the northern part of the country in 2011.

Findings from this study revealed that training on NASG were significantly associated with the utilization of NASG. This finding is in line with studies done in Western Nigeria, and Jimma Ethiopia [23, 33]. This could be because the refreshment and subject-specific training

**Table 4. Attitudes towards NASG utilization among obstetric care providers working in public health facilitates of southern Ethiopia, 2020.**

| Variables | Agree No (%) | Neutral | Disagree No (%) |
|---|---|---|---|
| The use of anti-shock garment (NASG) is unnecessary especially in the center where there is a facility for blood transfusion | 172 (41.8) | 42 (10.2) | 198 (48) |
| There is no need for NASG, since it is not readily available | 145 (35.2) | 49 (11.9) | 218 (52.9) |
| The garment (NASG) is expensive, therefore not affordable | 160 (38.8) | 47 (11.4) | 205 (49.7) |
| The NASG application and removal require a lot of procedures that take time. | 135 (32.8) | 39 (9.5) | 238 (57.8) |
| The NASG can transmit HIV to patients; hence it is not advisable to be used in a hospital setting | 217 (52.7) | 60 (14.6) | 135 (32.8) |
| A NASG is only beneficial to people in rural areas/primary care settings | 154 (37.4) | 43 (10.4) | 215 (52.2) |
| NASG is only meant to be utilized by doctors | 232 (56.4) | 49 (11.9) | 131 (31.8) |
| NASG is ineffective in patients with cervical lacerations | 184 (44.7) | 49 (11.9) | 179 (43.4) |

provided at facilities paved a way for obstetric care providers to acquire the necessary information which enabled them on how to use the non-pneumatic antishock garment. In addition, those who had training will have fresh memory about NASG which will enable them to apply the instrument when the need arises. This emphasizes the necessity of new hire training at each facility for new employees on life-saving instruments like NASG. A study conducted elsewhere also emphasized the importance of training on job performance and also enhancing positive attitude [36]. Moreover, the provision of training is also positively associated with the satisfaction of professionals and the need to apply the new knowledge that has been acquired, it will help the professionals to believe that they have improved their professional competence and the quality of health care that they can provide [37].

**Table 5. Utilization of non-pneumatic anti-shock garment among obstetric care providers working in public health facilities of southern Ethiopia, 2020.**

| Variables | Frequency | Percentage |
|---|---|---|
| **Used NASG for the management of PPH (N = 412)** | | |
| Yes | 200 | 48.54 |
| No | 212 | 51.46 |
| **Reason for non-utilization***(N = 212) | | |
| Didn't know it was available | 59 | 28.23 |
| No patient needed it | 37 | 17.70 |
| Don't know how to use it | 115 | 54.07 |
| **Do you use NASG every time there is PPH(N = 200)** | | |
| Yes | 109 | 54.50 |
| No | 91 | 45.50 |
| **If not, when do you use it***(N = 91) | | |
| severe pph | 54 | 59.34 |
| shock | 20 | 21.98 |
| when other method fail | 17 | 18.68 |

**Table 6. Factors associated with the utilization of non-pneumatic antishock garment among obstetric care providers in public health facilities of Southern Ethiopia, 2020.**

| Variable | Utilization of NASG | | COR (95%. C.I), P. Value | AOR (95% C.I), P-Value |
|---|---|---|---|---|
| | Yes | No | | |
| **Age** | | | | |
| 20–24 | 40 | 41 | 0.8 (0.4, 1.3), 0.4 | 0.9 (0.4, 2.1), 0.8 |
| 25–30 | 90 | 116 | 0.6 (0.4, 0.9), 0.03 | 0.7 (0.4, 1.2), 0.2 |
| >= 31 | 70 | 55 | 1 | 1 |
| **Year of experience in work** | | | | |
| 1–5 years | 105 | 136 | 1 | 1 |
| 6–10 years | 80 | 65 | 1.6 (1.1, 2.4), 0.03 | 1.3 (0.7, 2.1), 0.4 |
| 11 years and above | 15 | 11 | 1.8 (0.8, 4.0), 0.2 | 1.06(0.5, 3.7), 0.9 |
| **Trend of staff motivation** | | | | |
| Yes | 81 | 56 | 1.9 (1.3, 2.8), 0.003 | 1.22 (0.71, 1.9), 0.5 |
| No | 119 | 156 | 1 | 1 |
| **Training on NASG** | | | | |
| Yes | 108 | 42 | 4.8 (3.1, 7.4), 0.001 | 2.9 (1.7, 4.2), 0.000* |
| No | 92 | 170 | 1 | 1 |
| **Protocols about NASG** | | | | |
| Yes | 148 | 106 | 2.9 (1.9, 4.3), 0.000 | 1.3 (0.7, 2.1), 0.4 |
| No | 52 | 106 | 1 | 1 |
| **Attitude about NASG** | | | | |
| Agree | 96 | 67 | 1 | 1 |
| Neutral | 54 | 32 | 0.4 (0.2, 0.6), 0.001 | 0.4(0.2, 0.7), 0.003* |
| Disagree | 64 | 101 | 0.5 (0.3, 0.7), 0.000 | 0.4 (0.2, 0.7), 0.001* |
| **Type of facility** | | | | |
| Hospital | 162 | 131 | 2.6 (1.7, 4.1), 0.000 | 1.8 (1.0, 3.1), 0.034* |
| Health center | 38 | 81 | 1 | 1 |
| **Knowledge on NASG** | | | | |
| Good knowledge | 165 | 138 | 2.6 (1.6, 4.17), 0.000 | 2 (1.7, 3.4), 0.012* |
| Poor knowledge | 33 | 72 | 1 | 1 |

*Significant at P < 0.05.

In this study, obstetric care provider's attitude was significantly associated with their utilization of NASG. Compared with providers with a positive attitude towards the NASG, providers with a negative and neutral attitude were 59% and 61% less likely to use the NASG, respectively. This finding is in line with studies conducted in two regions of Ethiopia [33, 35]. The possible explanation for this could be, positive attitude towards the instrument might be related to an interest in knowing and exploring the instrument which in turn would enable the obstetric care providers to use the instrument when the demand arises. In addition, the neutral or undecided attitude of obstetric providers may indicate that they may be inadequately informed about the use of NASG. This is important for programmers in this particular issue, it will be an entry into the knowledge of the range of uninformed obstetric care providers in the health care facilities. In contrary to this finding, a study from Ondo State Nigeria [31] reported a non-significant relationship between attitude towards NASG and professionals utilization of NASG. The possible reason for this could be due to methodological differences (having different sampling methods and procedures, the difference in the study setting, and the difference in computation of the attitude variable).

The current study revealed that knowledge about NASG is significantly associated with the utilization of NASG. This is consistent with studies conducted in Ondo State Nigeria, Western Nigeria, Northern Ethiopia, and Jimma Ethiopia [23, 31, 33, 35]. This could be due to the fact that the more knowledge the obstetric care providers have, the more confidence they will have to apply the NASG. Moreover, the use of the instrument is composed of different steps and precautions, so that those who had adequate knowledge will be able to use it efficiently and effectively. In contrary to the current study, findings from Bayelsa state Nigeria stated a non-significant association between professional's knowledge and utilization of NASG [22]. The discrepancy could be due to the different types of obstetric care providers included in the current study, whereas only midwives were included in the other study. The other reason might be the difference in sample size used (the sample size in the current study is four-fold compared to this study conducted in Nigeria).

The current study revealed that facility type is significantly associated with the utilization of NASG. In countries like Ethiopia with limited resources, the distribution of basic facilities including NASG may not be even in hospitals and health centers. As was identified in the current study hospitals are more likely to use NASG than health centers. However, considering the delays and difficulties in accessing definitive care for postpartum hemorrhage management in different parts of the country, health centers should be targeted and addressed about the use of the NASG.

In this study, the year of experience, and age of the participants were not significantly associated with the use of NASG. The finding of this result is in agreement with the studies elsewhere [22, 23, 35].

The public health importance of this study is; postpartum hemorrhage, which is a leading cause of maternal death in developing countries including Ethiopia, its complications largely occur within a very narrow time window, lending themselves to very focused and targeted interventions. Successful management of postpartum hemorrhage will require a combination of approaches to expand access to skilled care and, at the same time, extend life-saving interventions. In low-resource settings, obstetric care providers have few resources with which to stabilize women with severe PPH. NASG reduces maternal mortality and morbidity by buying time and stabilizing women during delays in transport and receiving appropriate care. Knowledge of the NASG as a unique first-aid device with which to stabilize patients within the facility or while awaiting transport to a referral facility provides an increased ability to stabilize women in shock in primary care settings. With training obstetric care providers can stabilize hemorrhaging women with the NASG, preventing complications from postpartum hemorrhage. Therefore, determining the utilization status of NASG and different factors will enable the public policy to enforce the availability and utilization of the lifesaving and inexpensive instrument, thereby preventing devastating and life-threatening complications of postpartum hemorrhage.

## Conclusions and recommendations

In the current study, roughly half of the providers use the NASG to prevent complications from postpartum hemorrhage. Training on NASG, type of facility, knowledge, and attitude on NASG were significantly associated with obstetric care provider's utilization of NASG. Strategies and program initiatives should focus on strengthening in-service and continuous professional development training, thereby filling the knowledge and attitude gap among obstetric care providers. Health centers, which are mostly accessible by the majority of the rural Ethiopian community should be targeted in future programs for accessibility and utilization of NASG. Stakeholders and non-governmental organizations that are working on improving

maternal health could use this opportunity to increase the accessibility of NASG, and build the capacity of obstetric care providers.

## Supporting information

**S1 File. Data collection tool used for the study.**
(DOCX)

**S1 Dataset. Data set of the study.**
(DTA)

## Acknowledgments

The authors are grateful for the study participants and the data collectors for their voluntariness in the data collection process. We are grateful for personnel at hospitals and health centers who have supported us with the facilitation of the data collection.

## Author Contributions

**Conceptualization:** Yordanos Gizachew Yeshitila.

**Data curation:** Yordanos Gizachew Yeshitila.

**Formal analysis:** Yordanos Gizachew Yeshitila.

**Funding acquisition:** Yordanos Gizachew Yeshitila.

**Investigation:** Yordanos Gizachew Yeshitila.

**Methodology:** Yordanos Gizachew Yeshitila, Agegnehu Bante, Zeleke Aschalew, Bezawit Afework, Selamawit Gebeyehu.

**Project administration:** Yordanos Gizachew Yeshitila, Agegnehu Bante, Zeleke Aschalew, Bezawit Afework, Selamawit Gebeyehu.

**Resources:** Yordanos Gizachew Yeshitila, Agegnehu Bante.

**Software:** Yordanos Gizachew Yeshitila.

**Supervision:** Yordanos Gizachew Yeshitila.

**Validation:** Yordanos Gizachew Yeshitila, Agegnehu Bante, Zeleke Aschalew, Bezawit Afework, Selamawit Gebeyehu.

**Visualization:** Yordanos Gizachew Yeshitila, Agegnehu Bante, Zeleke Aschalew, Selamawit Gebeyehu.

**Writing – original draft:** Yordanos Gizachew Yeshitila.

**Writing – review & editing:** Yordanos Gizachew Yeshitila, Agegnehu Bante, Zeleke Aschalew, Bezawit Afework, Selamawit Gebeyehu.

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
