## [Decision Letter · Decision Letter 0]

27 Jul 2021

PONE-D-21-10132

Utilization of non-pneumatic anti-shock garment and associated factors for postpartum hemorrhage management among obstetric  care providers in public health facilities  of southern Ethiopia, 2020

PLOS ONE

Dear Dr. Yeshitila,

Thank you for submitting your manuscript to PLOS ONE. After careful consideration, we feel that it has merit but does not fully meet PLOS ONE’s publication criteria as it currently stands. Therefore, we invite you to submit a revised version of the manuscript that addresses the points raised during the review process.

A rebuttal letter that responds to each point raised by the academic editor and reviewer(s). You should upload this letter as a separate file labeled 'Response to Reviewers'.A marked-up copy of your manuscript that highlights changes made to the original version. You should upload this as a separate file labeled 'Revised Manuscript with Track Changes'.An unmarked version of your revised paper without tracked changes. You should upload this as a separate file labeled 'Manuscript'

We look forward to receiving your revised manuscript.

Kind regards,

Catherine S. Todd

Academic Editor

PLOS ONE

Journal Requirements:

2**. **We note that the grant information you provided in the ‘Funding Information’ and ‘Financial Disclosure’ sections do not match. When you resubmit, please ensure that you provide the correct grant numbers for the awards you received for your study in the ‘Funding Information’ section.

**3****.**  In your Data Availability statement, you have not specified where the minimal data set underlying the results described in your manuscript can be found. PLOS defines a study's minimal data set as the underlying data used to reach the conclusions drawn in the manuscript and any additional data required to replicate the reported study findings in their entirety. All PLOS journals require that the minimal data set be made fully available. For more information about our data policy, please see http://journals.plos.org/plosone/s/data-availability. "

Additional Editor Comments (if provided):

I appreciate the authors' work on this paper and the findings are interesting. As noted by the reviewers, a number of revisions are needed as well as editing for language and grammar. Please also make the data available to ensure compliance with PLoS One's open data policy.

Reviewers' comments:

Reviewer's Responses to Questions

**Comments to the Author**

1. Is the manuscript technically sound, and do the data support the conclusions?

Reviewer #1: Partly

Reviewer #2: Yes

2. Has the statistical analysis been performed appropriately and rigorously? 

Reviewer #1: Yes

Reviewer #2: Yes

3. Have the authors made all data underlying the findings in their manuscript fully available?

Reviewer #1: No

Reviewer #2: Yes

4. Is the manuscript presented in an intelligible fashion and written in standard English?

Reviewer #1: Yes

Reviewer #2: No

5. Review Comments to the Author

Reviewer #1: Overall comment:

The aims of this study are valuable, and the authors’ recommendations for increasing the use of NASG by targeting primary health centers are a valuable message. However there are some issues in the reporting of the results and conclusions that should be clarified to reflect the data. In particular, the conclusion should recognize that the Ethiopian government and CHAI have achieved a significant victory if within a random sample of providers, nearly half are using the NASG for management of hypovolemic shock and PPH. Several other countries are working towards this goal but are not close to this achievement. Attention should also be paid to the interpretation of the scaled attitude data. The authors should be attentive to reporting this data in an objective manner. Otherwise, the authors’ conclusions appear sound and provide important feedback about the introduction of the NASG in Ethiopia.

There are some specific language edits and clarifications of the data that are needed for this manuscript to be ready for publication, which I have detailed below.

1. The data for this study is indicated as being fully available according to PLOS One’s guidelines. However some data has been simplified for analysis and is not available in its original form within the supplemental material. I have provided suggestions for how this could be corrected within the manuscript in comment #9.

2. Language:

a. Please correct the spelling of “professionals” in the abstract and throughout the text.

b. Please correct “settlement” to “establishment” in the abstract and throughout the text.

c. “nasg” should be capitalized NASG as an acronym throughout the text.

d. Authors should check spelling and capitalization throughout the text.

3. Line 40 “A significant proportion of obstetric care providers do not use non pneumatic antishock garment for preventing complications from postpartum hemorrhage.”

This is a subjective and possibly misleading statement, as the authors do not give a reference point for comparison. It would be more accurate to say “roughly half of providers are using the NASG,” and then go on to describe the reasons the authors would recommend that actions be taken to increase this proportion.

4. Line 81. Add period: “excessive bleeding. With the introduction…”

5. Lines 103-105 “Most the studies conducted before used a small sample size (60-112) [19-22] and in developed countries where maternal mortality is less than 64/100 000 [23-25]."

If the authors are referring to implementation studies (studies examining utilization), both parts of this statement are not accurate. Please refer to Mbaruku study was conducted with a sample size of 1,713 women in Tanzania, published in 2018.

https://pubmed.ncbi.nlm.nih.gov/30340602/

6. Line 234-235 Regarding the use of the NASG on patients with a viable fetus. This is perhaps not worth mentioning in this paper as this issue is not being explored in-depth. Conclusive studies have not been done to investigate safety of the NASG with a viable fetus, and the NASG can be considered for use when “there is no other way to save the mother’s life and both mother and fetus will die.” (https://www.nqocncop.org/forum/welcome-to-the-forum/complementary-role-of-non-pneumatic-anti-shock-garment-nasg-in-management-hemorrhagic-shock-in-obstetrics)

7. Line 238: Table 3

For the knowledge questions, the authors could consider indicating all the correct responses for the readers who may not have this information.

8. Line 248 “Over all, 245 (59.5%) of the respondts had unfavourbale attitude towards non neumatic antishock garment.”

This statement appears to be attempting to summarize the results of the 8 scaled attitude statements, and does not appear to be a sound conclusion. It's not clear how the number 245 was derived. If I understand the author’s interpretation, this statistic counts neutral responses as “unfavorable” rather than neutral. Please see the next comment for feedback on the interpretation of the Likert scale responses. This statement should be removed and replaced with information that is firmly rooted in the data, however I am not able to provide more guidance as not enough information about the attitude responses has been made available in the manuscript.

9. Table 4: Attitudes

Reporting Likert scale data can pose a challenge. The 5 point Likert scale used here allowed respondents to report a neutral or undecided response. However the authors have chosen to collapse the 5 possible responses into a binary variable, “favorable” or “unfavorable”. Collapsing 5 responses without differentiation is problematic. Characterizing neutral responses as negative can mislead the reader and impacts the authors’ interpretation.

“Undecided” responses are important within this context as they indicate the number of respondents who may feel insufficiently informed on the NASG. A respondent who feels uninformed about the NASG should be considered substantively different from a respondent who feels informed and has formed a negative attitude. The neutral response is in line with the “I don’t know” categories in Table 3. It is relevant and important for those who are programming activities on the NASG to know the scope of uniformed staff within the health facilities, and this information supports the authors’ conclusion that further training is required.

This data should be clearly presented to the readers. The authors should add at least 1 column to this table to show the neutral responses, or alter this table to include 5 columns, to reflect all possible responses to gain the full benefit of the Likert scale. Including the “strongly agrees” and “strongly disagrees” would add value to the authors ‘ work. This data can be represented in a table or a graph displaying all percentages. An example of clear representation of 5-point responses can be found in this article:

Haffer H, Schömig F, Rickert M, Randau T, Raschke M, Wirtz D, Pumberger M, Perka C. Impact of the COVID-19 Pandemic on Orthopaedic and Trauma Surgery in University Hospitals in Germany: Results of a Nationwide Survey. J Bone Joint Surg Am. 2020 Jul 15;102(14):e78. doi: 10.2106/JBJS.20.00756. PMID: 32675666; PMCID: PMC7431148.

In addition, I would recommend that “favorable” be changed to “disagree” and “unfavorable” be changed to “Agree” which are the standard convention in English.

These changes should ensure that readers do not misinterpret the important results in this table. These changes should also be reflected in Table 6, and discussed in the results.

10. Line 284 “substantiate” is not correct in this context. Sentence could read “The NASG has been proven effective as a tool to stabilize….”

11. Line 285 Replace “absolute” with “definitive.”

12. Line 311 “inception” The authors’ intention with this word is not clear. Possibly “initial” or “new hire trainings”?

13. Line 365 “A significant proportion of obstetric care providers do not use non pneumatic antishock garment for preventing complications from postpartum hemorrhage.”

Same correction recommended as line 40.

Reviewer #2: Please correct the grammar and some spelling errors in the manuscript. Please mention whether all the obstetric care workers in Southern Ethiopia have previously been offered training on how the NASG is used. Also mention whether refresher trainings are available. Also indicate whether the NASG is actually available at the institutions where the study was done. On page 13 please indicate the type of random sampling that was used to select the institutions that participated in the study. The title of Table 1 gives the impression that these are the socio-demographic characteristics of all obstetric care workers in the region. Better to title the table to indicate that these are the socio-demographic characteristics of only the obstetric care workers who participated in the study. On Table 2 under ''Professions'' what does ''Emergency surgery'' refer to? Is it referring to general surgeons? On page 18 (lines 229 to 231) what denominator was used to determine the proportion of respondents that knew that the NASG is made up of six parts? On Table 3 (page 19) was the third row from the bottom meant to say ''Transport to other facilities''? The type of questions in Table 4 and Table 5 ideally should not be asked of someone who says that they have never heard of the NASG. The denominators should exclude those who say that they have never heard of the NASG.

6. PLOS authors have the option to publish the peer review history of their article (what does this mean?). If published, this will include your full peer review and any attached files.

Reviewer #1: **Yes: **Michelle Skaer Therrien

Reviewer #2: No

---

## [Author Response · Author response to Decision Letter 0]

4 Aug 2021

Title: Utilization of non-pneumatic anti-shock garment and associated factors for postpartum hemorrhage management among obstetric care providers in public health facilities of southern Ethiopia, 2020

Authors:

Yordanos Gizachew Yeshitila (yordanos.gizachew@yahoo.com)

Agegnehu Bante (agegnehubante@gmail.com)

Zeleke Aschalew (zelekeaschalew@gmail.com)

Bezawit Afework (bezawitafework2010@gmail.com)

Selamawit Gebeyehu (emugebe.sg@gmail.com)

Responses compiled by: Yordanos Gizachew Yeshitila 

On behalf of the authors

Responses to editor’s comment 

Editor’s comments, suggestions and question are considered and carefully revised the manuscript as per the suggestions and comments. Thank you for the feedback. We have addressed each comment below and within the revised manuscript (tracked changes and clean version).

Sincerely,

Yordanos Gizachew Yeshitila , on behalf of the authors

Editors comment 

Item 1:

An unmarked version of your revised paper without tracked changes. You should upload this as a separate file labeled 'Manuscript'

Response 1: Thank you dear editor, we have included all the three requested items

Item 2: When submitting your revision, we need you to address these additional requirements.

Response 2: Dear editor, we have gone through the manuscript, and revised it according to the PLOS ONE's style requirement 

Item 3: We note that the grant information you provided in the ‘Funding Information’ and ‘Financial Disclosure’ sections do not match. When you resubmit, please ensure that you provide the correct grant numbers for the awards you received for your study in the ‘Funding Information’ section.

Response3: Dear editor, we have provided correct grant numbers for the awards we received for our study in the ‘Funding Information’ section on the electronic editorial manger section as well as on the manuscript. (However, our institution is not listed among the funders in the dropdown menu)

Item 4: in your Data Availability statement, you have not specified where the minimal data set underlying the results described in your manuscript can be found. PLOS defines a study's minimal data set as the underlying data used to reach the conclusions drawn in the manuscript and any additional data required to replicate the reported study findings in their entirety. All PLOS journals require that the minimal data set be made fully available. For more information about our data policy, please see http://journals.plos.org/plosone/s/data-availability.

Response 4: Dear editor, we have provided the data availability statement as per the journal requirement on the electronic editorial manger data availability section

Item 5: Upon re-submitting your revised manuscript, please upload your study’s minimal underlying data set as either Supporting Information files or to a stable, public repository and include the relevant URLs, DOIs, or accession numbers within your revised cover letter. For a list of acceptable repositories, please see http://journals.plos.org/plosone/s/data-availability#loc-recommended-repositories. Any potentially identifying patient information must be fully anonymized.

Response 5: Dear editor, we have uploaded study’s minimal underlying data set as Supporting Information files

Item 6: Please include captions for your Supporting Information files at the end of your manuscript, and update any in-text citations to match accordingly. Please see our Supporting Information guidelines for more information: http://journals.plos.org/plosone/s/supporting-information.

Response 6: Dear editor, we have included captions for our Supporting Information files at the end of our manuscript, and updated any in-text citations to match accordingly

Item 7: I appreciate the authors' work on this paper and the findings are interesting. As noted by the reviewers, a number of revisions are needed as well as editing for language and grammar. Please also make the data available to ensure compliance with PLoS One's open data policy.

Response 7: Dear editor, thank you so much. We have gone through each and every concern and comments raised, and responded accordingly. We have also exhaustively worked on the editing for language and grammar

Reponses to reviewers 

Reviewer’s comments and suggestions are considered and carefully revised the manuscript as per the suggestions and comments. I have addressed the requests and concerns raised by both the reviewers. I am very thankful to the potential reviewers for suggestions and comments, which substantially improved the manuscript

Response to reviewer 1 

Item 1: The aims of this study are valuable, and the authors’ recommendations for increasing the use of NASG by targeting primary health centers are a valuable message. However there are some issues in the reporting of the results and conclusions that should be clarified to reflect the data. In particular, the conclusion should recognize that the Ethiopian government and CHAI have achieved a significant victory if within a random sample of providers, nearly half are using the NASG for management of hypovolemic shock and PPH. Several other countries are working towards this goal but are not close to this achievement. Attention should also be paid to the interpretation of the scaled attitude data. The authors should be attentive to reporting this data in an objective manner. Otherwise, the authors’ conclusions appear sound and provide important feedback about the introduction of the NASG in Ethiopia

Response 1: Dear reviewer, thank you for your insights and thank you so much for meticulous observation and comments which significantly improved the manuscript.

*We have considered and incorporated your suggestion on the interpretation of the scaled attitude data.

Item 2: The data for this study is indicated as being fully available according to PLOS One’s guidelines. However some data has been simplified for analysis and is not available in its original form within the supplemental material. I have provided suggestions for how this could be corrected within the manuscript in comment #9.

Response 2: Dear reviewer, we have provided the data as per your request. 

(Page 14-15)

Item 3: Language:

a. Please correct the spelling of “professionals” in the abstract and throughout the text.

b. Please correct “settlement” to “establishment” in the abstract and throughout the text.

c. “nasg” should be capitalized NASG as an acronym throughout the text.

d. Authors should check spelling and capitalization throughout the text.

Response 3: Dear reviewer, we truly appreciate your suggestions and comments. We have incorporated each of them in the revised manuscript. The changes we made are presented on the revised manuscript with track change.

Item 4: Line 40 “A significant proportion of obstetric care providers do not use non pneumatic antishock garment for preventing complications from postpartum hemorrhage.”

This is a subjective and possibly misleading statement, as the authors do not give a reference point for comparison. It would be more accurate to say “roughly half of providers are using the NASG,” and then go on to describe the reasons the authors would recommend that actions be taken to increase this proportion

Response 4: Dear reviewer, we have corrected the sentences as per your suggestion. 

Page 2, line 44-45 and page21, line 373-374

Item 5: Line 81. Add period: “excessive bleeding. With the introduction…”

Response 5: Dear reviewer, we have made the correction. Page 3, line 82

Item 6: Lines 103-105 “Most the studies conducted before used a small sample size (60-112) [19-22] and in developed countries where maternal mortality is less than 64/100 000 [23-25]."

If the authors are referring to implementation studies (studies examining utilization), both parts of this statement are not accurate. Please refer to Mbaruku study was conducted with a sample size of 1,713 women in Tanzania, published in 2018.

https://pubmed.ncbi.nlm.nih.gov/30340602/

Response 6: Dear reviewer, thank you for thorough review of our work. 

The first part of the paragraph was intended to show those studies conducted on professionals knowledge and use of NASG, the references listed, and the sample size they used (in brackets) were Faiza et al (60), Kombo et al (100), Kolade et al (110), and Onasoga et al (112), however, for the second part of the statement as you have provided us with the references there are studies conducted in developing countries, where we were trying to show the studies from developed countries. So we removed the second part of the statement. 

Item 7: Line 234-235 Regarding the use of the NASG on patients with a viable fetus. This is perhaps not worth mentioning in this paper as this issue is not being explored in-depth. Conclusive studies have not been done to investigate safety of the NASG with a viable fetus, and the NASG can be considered for use when “there is no other way to save the mother’s life and both mother and fetus will die.” (https://www.nqocncop.org/forum/welcome-to-the-forum/complementary-role-of-non-pneumatic-anti-shock-garment-nasg-in-management-hemorrhagic-shock-in-obstetrics)

Response 7: Dear reviewer, thank you again for your meticulous observation. It has been stated there is no absolute contraindication for use of NASG for PPH, However, regarding the use of NASG, as an indication, it is stated on pathfinder’s guideline that (page 87) “The NASG could be used to manage any condition where there is severe bleeding below the diaphragm. Our studies have documented use with all forms of obstetric hemorrhage, as long as the fetus is not viable in utero”

We would appreciate if you take look at the guideline 

(https://www.pathfinder.org/publications/prevention-recognition-management-of-postpartum-hemorrhage-trainers-guide/

Item 8: For the knowledge questions, the authors could consider indicating all the correct responses for the readers who may not have this information.

Response 8: Dear reviewer, we have indicated the correct responses on the table with symbol and provided description for the symbols underneath the table.

Item 9: Line 248 “Over all, 245 (59.5%) of the respondts had unfavourbale attitude towards non neumatic antishock garment.”

This statement appears to be attempting to summarize the results of the 8 scaled attitude statements, and does not appear to be a sound conclusion. It's not clear how the number 245 was derived. If I understand the author’s interpretation, this statistic counts neutral responses as “unfavorable” rather than neutral. Please see the next comment for feedback on the interpretation of the Likert scale responses. This statement should be removed and replaced with information that is firmly rooted in the data, however I am not able to provide more guidance as not enough information about the attitude responses has been made available in the manuscript

Response 9: Dear reviewer, we are grateful for the comments and lesson on this specific issue, As you have emphasized, the neutral or undecided attitude of the professionals is another core aspect for future programmers and interventional projects on the area regarding NASG utilization. 

*We have provided data on the neutral or undecided attitude aspect of the professionals (Page 14)

*The whole analysis was changed and interpreted again accounting for the neutral attitude (Page 16-17)

* We have also included under discussion part (Page9, line 320-339

Item 10: Table 4: 

Attitudes

Reporting Likert scale data can pose a challenge. The 5 point Likert scale used here allowed respondents to report a neutral or undecided response. However the authors have chosen to collapse the 5 possible responses into a binary variable, “favorable” or “unfavorable”. Collapsing 5 responses without differentiation is problematic Characterizing neutral responses as negative can mislead the reader and impacts the authors’ interpretation.

“Undecided” responses are important within this context as they indicate the number of respondents who may feel insufficiently informed on the NASG. A respondent who feels uninformed about the NASG should be considered substantively different from a respondent who feels informed and has formed a negative attitude. The neutral response is in line with the “I don’t know” categories in Table 3. It is relevant and important for those who are programming activities on the NASG to know the scope of uniformed staff within the health facilities, and this information supports the authors’ conclusion that further training is required.

This data should be clearly presented to the readers. The authors should add at least 1 column to this table to show the neutral responses, or alter this table to include 5 columns, to reflect all possible responses to gain the full benefit of the Likert scale. Including the “strongly agrees” and “strongly disagrees” would add value to the authors ‘ work. This data can be represented in a table or a graph displaying all percentages. An example of clear representation of 5-point responses can be found in this article:

Haffer H, Schömig F, Rickert M, Randau T, Raschke M, Wirtz D, Pumberger M, Perka C. Impact of the COVID-19 Pandemic on Orthopaedic and Trauma Surgery in University Hospitals in Germany: Results of a Nationwide Survey. J Bone Joint Surg Am. 2020 Jul 15;102(14):e78. doi: 10.2106/JBJS.20.00756. PMID: 32675666; PMCID: PMC7431148.

In addition, I would recommend that “favorable” be changed to “disagree” and “unfavorable” be changed to “Agree” which are the standard convention in English.

These changes should ensure that readers do not misinterpret the important results in this table. These changes should also be reflected in Table 6, and discussed in the results.

Response 10: Thank you dear reviewer, 

We have reviewed the article you suggested 

(https://journals.lww.com/jbjsjournal/Fulltext/2020/07150/Impact_of_the_COVID_19_Pandemic_on_Orthopaedic_and.6.aspx

We have provided data on the neutral or undecided attitude aspect of the professionals (Page 14)

We have changed that “favorable” be changed to “agree” and “unfavorable” be changed to “disagree

Item 11: Line 284 “substantiate” is not correct in this context. Sentence could read “The NASG has been proven effective as a tool to stabilize….”

Response 11: Dear reviewer, we have revised it as per your suggestion, (Page 17, line 286)

Item 12: Line 285 Replace “absolute” with “definitive.”

Response 12: Dear reviewer, we have revised it as per your suggestion Page 17, line 287)

Item 13: Line 311 “inception” The authors’ intention with this word is not clear. Possibly “initial” or “new hire trainings”?

Response 13: Dear reviewer, the inception training were meant to be new hire training, as per your suggestion, we have corrected for clarity (Page 18, line 313)

Item 14: Line 365 “A significant proportion of obstetric care providers do not use non pneumatic antishock garment for preventing complications from postpartum hemorrhage.”

Same correction recommended as line 40.

Response 14: Dear reviewer, We have made the correction

Response to reviewer 2

Dear reviewer, thank you for your insights and thank you so much for meticulous observation and comments which significantly improved the manuscript.

Item 1: Please correct the grammar and some spelling errors in the manuscript.

Response 1: Dear reviewer, thank you. We have gone through the manuscript and made all the possible corrections. (changes can be found on the tracked version)

Item 2: Please mention whether all the obstetric care workers in Southern Ethiopia have previously been offered training on how the NASG is used.

Response 2: Dear reviewer, thank you. We have included obstetric care providers in health facilities where NASG is available, however weather they received training or not was one of the variable we assessed on table 2 (Page 10-11).

Item 3: Also mention whether refresher trainings are available

Response 3: Dear reviewer, same as to the above concern, we have included obstetric care providers in health facilities where NASG is available, however weather they received training or not was one of the variable we assessed on table 2 (Page 10-11).

Item 4: Also indicate whether the NASG is actually available at the institutions where the study was done.

Response 4: Dear reviewer, thank you. We have included obstetric care providers in health facilities where NASG is available, we have incorporated this under the method section, under subsection Study design and population (Page 6 line 132-133)

Item 5: On page 13 please indicate the type of random sampling that was used to select the institutions that participated in the study.

Response 5: Dear reviewer, thank you. We have indicated on page 6, line 146

Item 6: The title of Table 1 gives the impression that these are the socio-demographic characteristics of all obstetric care workers in the region. Better to title the table to indicate that these are the socio-demographic characteristics of only the obstetric care workers who participated in the study

Response 6: Dear reviewer, we truly appreciate your suggestion, we have revised the title as per your suggestion. (Page 9, line 210-211

Item 7: On Table 2 under ''Professions'' what does ''Emergency surgery'' refer to? Is it referring to general surgeons? 

Response 7: Dear reviewer, we apologize for using the short version of the title. The profession is Integrated Emergency Obstetric Surgery, not general surgeons. We would appreciate if you visit the link for better clarification (https://ethiopia.unfpa.org/en/news/msc-programme-integrated-emergency-obstetric-surgery-launched

Item 8: On page 18 (lines 229 to 231) what denominator was used to

Response 8: Dear reviewer, we have used the denominator of 338 (those who heard about NASG) we have included the denominator for each question.

Item 9: On Table 3 (page 19) was the third row from the bottom meant to say ''Transport to other facilities''?

Response 9: Dear reviewer, thank you. Yes, it is mean to say ''Transport to other facilities''?, we have modified on the table

Item 10: The type of questions in Table 4 and Table 5 ideally should not be asked of someone who says that they have never heard of the NASG. The denominators should exclude those who say that they have never heard of the NASG.

Response 10: Dear reviewer, we are so much grateful for the suggestion, which we believe clear confusion for future readers. We have included the denominator for each question.

---

## [Decision Letter · Decision Letter 1]

7 Sep 2021

PONE-D-21-10132R1Utilization of non-pneumatic anti-shock garment and associated factors for postpartum hemorrhage management among obstetric  care providers in public health facilities  of southern Ethiopia, 2020PLOS ONE

Dear Dr. Yeshitila,

Thank you for submitting your manuscript to PLOS ONE. After careful consideration, we feel that it has merit but does not fully meet PLOS ONE’s publication criteria as it currently stands. Therefore, we invite you to submit a revised version of the manuscript that addresses the points raised during the review process.

We look forward to receiving your revised manuscript.

Kind regards,

Catherine S. Todd

Academic Editor

PLOS ONE

Journal Requirements:

Additional Editor Comments:

My thanks to the authors for undertaking this revision, which have improved the manuscript. In addition to the remaining critiques from the reviewers, please ensure the de-identified data set is included with the next revision, as required by the journal.

Reviewers' comments:

Reviewer's Responses to Questions

**Comments to the Author**

1. If the authors have adequately addressed your comments raised in a previous round of review and you feel that this manuscript is now acceptable for publication, you may indicate that here to bypass the “Comments to the Author” section, enter your conflict of interest statement in the “Confidential to Editor” section, and submit your "Accept" recommendation.

Reviewer #1: (No Response)

Reviewer #2: (No Response)

2. Is the manuscript technically sound, and do the data support the conclusions?

Reviewer #1: Yes

Reviewer #2: Yes

3. Has the statistical analysis been performed appropriately and rigorously? 

Reviewer #1: Yes

Reviewer #2: Yes

4. Have the authors made all data underlying the findings in their manuscript fully available?

Reviewer #1: No

Reviewer #2: Yes

5. Is the manuscript presented in an intelligible fashion and written in standard English?

Reviewer #1: Yes

Reviewer #2: No

6. Review Comments to the Author

Reviewer #1: This revision has substantially addressed the issues that were raised in my comments as well as those of the other reviewer. I would like to thank the authors for their work to strengthen this paper. I have only minor corrections to recommends, otherwise I approve this paper.

1. Authors indicate in reply to the editors that the minimal data set is in the supporting information (Response 5 to the editor). In reviewing the supporting information, I noted only the tool used to collect the data, but did not find the original data. I believe this may remain to be addressed.

2. A comment: thank you to the authors for clarifying the reference on use with a viable fetus. Please note the Pathfinder document used as a reference is based on information from 2007, and has not been updated. It is considered out-of-date in light of several years of experience with the NASG in the field.

3. p.12 Velcro should have an asterisk as well, as the NASG is made of neoprene and velcro.

4. Line 238/239 The number is 198 and percentage is 48% of respondents disagreeing with the statement according to the table.

5. Line 240. The percentage is 39%, not 61% according to the table.

6. Line 295 "magnitude" (spelling correction)

Reviewer #2: There are still some concerns with the grammar. Please attend to the grammar or spelling errors in the following lines of the manuscript - 63, 67, 73, 75, 83, 95, 96 to 98 100, 105, 116, 131, 151, 159 to 160, 191, 198, 199, 207, 264, 279 and 373. Line 141 says the final sample size was 422 mothers? I think this is meant to read 422 health workers.

7. PLOS authors have the option to publish the peer review history of their article (what does this mean?). If published, this will include your full peer review and any attached files.

Reviewer #1: No

Reviewer #2: No

---

## [Author Response · Author response to Decision Letter 1]

9 Sep 2021

Responses to editor’s comment 

Dear editor, Thank you for the feedback. We have addressed comments and suggestions below and within the revised manuscript (tracked changes and clean version).

Sincerely,

Yordanos Gizachew Yeshitila, on behalf of the authors

An unmarked version of your revised paper without tracked changes. You should upload this as a separate file labeled 'Manuscript'

**Thank you dear editor, we have included all the three requested items

**Dear editor, thank you, we have gone tough the reference and made correction, we have showed the changes we made with highlight on the track version of the manuscript. 

**We have also mentioned the changes we made to reference based on the suggestion from one of the reviewer on the cover letter

My thanks to the authors for undertaking this revision, which have improved the manuscript. In addition to the remaining critiques from the reviewers, please ensure the de-identified data set is included with the next revision, as required by the journal.

** Dear editor, we are grateful for the feedbacks. We have uploaded the de-identified data set which is used for the study. 

Responses to the Reviewers' comment and question 

Reviewer’s comments and suggestions are carefully considered and revised the manuscript as per the suggestions and comments. I have incorporated the suggestions and comments raised by both the reviewers. I am very thankful to the potential reviewers for suggestions and comments, which substantially improved the manuscript

Reviewer 1

Item 1: This revision has substantially addressed the issues that were raised in my comments as well as those of the other reviewer. I would like to thank the authors for their work to strengthen this paper. I have only minor corrections to recommends, otherwise I approve this paper

Response 1: Thank you dear reviewer for the feedback. We have incorporated your suggestion while revising the manuscript.

Item 2: Authors indicate in reply to the editors that the minimal data set is in the supporting information (Response 5 to the editor). In reviewing the supporting information, I noted only the tool used to collect the data, but did not find the original data. I believe this may remain to be addressed.

Response 2: Dear reviewer, we have uploaded the de-identified data set as per the inquiry. We apologize for not addressing the issue last time.

Item 3: A comment: thank you to the authors for clarifying the reference on use with a viable fetus. Please note the Pathfinder document used as a reference is based on information from 2007, and has not been updated. It is considered out-of-date in light of several years of experience with the NASG in the field.

Response 3: Dear reviewer, we truly appreciate your suggestions, for this specific inquiry we have replaced the reference with (Downing, J. et al) , DOI 10.1186/s12913-015-0694-6) although we didn’t mention it during our response last time. In addition please be informed that we have use other references together with pathfinder while developing the data collection tool including FIGO.

Item 4: . p.12 Velcro should have an asterisk as well, as the NASG is made of neoprene and velcro.

Response 4: Dear reviewer, we have corrected the sentences as per your suggestion. (Table 3)

Item 5: Line 238/239 The number is 198 and percentage is 48% of respondents disagreeing with the statement according to the table.

Response 5: Dear reviewer, thank you, we have made the correction. Page 14, line 245-246

Item 6: Line 240. The percentage is 39%, not 61% according to the table.

Response 6: Dear reviewer, thank you for your thorough review of our work. We have made the correction. Page 14, line 248-249

Item 7: Line 295 "magnitude" (spelling correction)

Response 7: Dear reviewer, thank you for thorough review of our work. We have made the correction. Page 18, line 303

Reviewer 2 

Item 1: There are still some concerns with the grammar. Please attend to the grammar or spelling errors in the following lines of the manuscript - 63, 67, 73, 75, 83, 95, 96 to 98 100, 105, 116, 131, 151, 159 to 160, 191, 198, 199, 207, 264, 279 and 373.

Response 1: Dear reviewer, thank you. We have gone through the manuscript and made the correction as you pointed them out for us. We have made the changes on the track version of the manuscript

Item 2: Line 141 says the final sample size was 422 mothers? I think this is meant to read 422 health workers.

Response 2: Thank you dear reviewer, yes it health workers, we have made the correction accordingly.

---

## [Editor Report · Decision Letter 2]

15 Sep 2021

PONE-D-21-10132R2Utilization of non-pneumatic anti-shock garment and associated factors for postpartum hemorrhage management among obstetric  care providers in public health facilities  of southern Ethiopia, 2020PLOS ONE

Dear Dr. Yeshitila,

Thank you for submitting your manuscript to PLOS ONE. After careful consideration, we feel that it has merit but does not fully meet PLOS ONE’s publication criteria as it currently stands. Therefore, we invite you to submit a revised version of the manuscript that addresses the points raised during the review process. Please submit your revised manuscript by Oct 30 2021 11:59PM. If you will need more time than this to complete your revisions, please reply to this message or contact the journal office at plosone@plos.org. Please include the following items when submitting your revised manuscript:A rebuttal letter that responds to each point raised by the academic editor and reviewer(s). You should upload this letter as a separate file labeled 'Response to Reviewers'.A marked-up copy of your manuscript that highlights changes made to the original version. You should upload this as a separate file labeled 'Revised Manuscript with Track Changes'.An unmarked version of your revised paper without tracked changes. You should upload this as a separate file labeled 'Manuscript'.If applicable, we recommend that you deposit your laboratory protocols in protocols.io to enhance the reproducibility of your results. Protocols.io assigns your protocol its own identifier (DOI) so that it can be cited independently in the future. For instructions see: https://journals.plos.org/plosone/s/submission-guidelines#loc-laboratory-protocols. Additionally, PLOS ONE offers an option for publishing peer-reviewed Lab Protocol articles, which describe protocols hosted on protocols.io. Read more information on sharing protocols at https://plos.org/protocols?utm_medium=editorial-email&utm_source=authorletters&utm_campaign=protocols.

We look forward to receiving your revised manuscript.

Kind regards,

Catherine S. Todd

Academic Editor

PLOS ONE

Journal Requirements:

Additional Editor Comments (if provided):

I thank the authors for their efforts to respond to reviewer comments. Despite this manuscript being revised twice, there remain considerable grammatical, formatting, and spelling errors that need to be addressed and corrected. Further, the non-pneumatic anti-shock garment (NASG) should be spelled out once and then the abbreviation used consistently through the remainder of the document. The manuscript would also greatly benefit from being made more concise - for example, the Introduction section states in three different places that postpartum hemorrhage is a leading cause of maternal mortality and gives the reasons for delayed care that lead to mortality. Please summarize this in one paragraph, describe the NASG in the next paragraph, discuss the introduction and use of NASG in Ethiopia briefly in a third paragraph and then state the gap in the evidence and the reason for the paper in the last paragraph. The Results section can similarly be abbreviated by removing double-reporting of proportions in the text that are also present in the tables. Simply summarize the most remarkable finding or two findings from each table and data not portrayed in tables. Please streamline the tables by only using one decimal place. Last, please streamline the Discussion section - the results should only be contextualized with the first paragraph summarizing the main findings. The current first paragraph re-states the Introduction - please remove this paragraph and simply summarize the main findings. Please take the time to carefully revise the manuscript before re-submission and consider engaging a professional editor who is a native English speaker.
---

## [Author Response · Author response to Decision Letter 2]

23 Sep 2021

Item 1: I thank the authors for their efforts to respond to reviewer comments. Despite this manuscript being revised twice, there remain considerable grammatical, formatting, and spelling errors that need to be addressed and corrected. Further, the non-pneumatic anti-shock garment (NASG) should be spelled out once and then the abbreviation used consistently through the remainder of the document. The manuscript would also greatly benefit from being made more concise - for example, the Introduction section states in three different places that postpartum hemorrhage is a leading cause of maternal mortality and gives the reasons for delayed care that lead to mortality. Please summarize this in one paragraph, describe the NASG in the next paragraph, discuss the introduction and use of NASG in Ethiopia briefly in a third paragraph, and then state the gap in the evidence and the reason for the paper in the last paragraph. The Results section can similarly be abbreviated by removing double-reporting of proportions in the text that are also present in the tables. Simply summarize the most remarkable finding or two findings from each table and data not portrayed in tables. Please streamline the tables by only using one decimal place. Last, please streamline the Discussion section - the results should only be contextualized with the first paragraph summarizing the main findings. The current first paragraph re-states the Introduction - please remove this paragraph and simply summarize the main findings. Please take the time to carefully revise the manuscript before re-submission and consider engaging a professional editor who is a native English speaker.

Response 1: Dear editor, we are very much grateful for the suggestions and comments which potentially improved our work. 

• We have made all the corrections as per your suggestion (please kindly see the revised version with track change )

• We have exhaustively reviewed the manuscript and according to your recommendation, we engaged a professional editor (the changes made and are shown the revised manuscript with track change).

---

## [Editor Report · Decision Letter 3]

6 Oct 2021

Utilization of non-pneumatic anti-shock garment and associated factors for postpartum hemorrhage management among obstetric  care providers in public health facilities  of southern Ethiopia, 2020

PONE-D-21-10132R3

Dear Dr. Yeshitila,

We’re pleased to inform you that your manuscript has been judged scientifically suitable for publication and will be formally accepted for publication once it meets all outstanding technical requirements.

Kind regards,

Catherine S. Todd

Academic Editor

PLOS ONE

Additional Editor Comments (optional):

This manuscript is much improved and I thank the authors for their perseverance and attention to detail.
---

## [Editor Report · Acceptance letter]

8 Oct 2021

PONE-D-21-10132R3 

Utilization of non-pneumatic anti-shock garment and associated factors for postpartum hemorrhage management among obstetric care providers in public health facilities of southern Ethiopia, 2020 

Dear Dr. Yeshitila:

I'm pleased to inform you that your manuscript has been deemed suitable for publication in PLOS ONE. Congratulations! Your manuscript is now with our production department. 

Kind regards, 

on behalf of

Dr. Catherine S. Todd 

Academic Editor

PLOS ONE